# Do Compactness and Poly-Centricity Mitigate PM_10_ Emissions? Evidence from Yangtze River Delta Area

**DOI:** 10.3390/ijerph16214204

**Published:** 2019-10-30

**Authors:** Jing Tao, Ying Wang, Rong Wang, Chuanmin Mi

**Affiliations:** 1College of Economics and Management, Nanjing University of Aeronautics and Astronautics, Nanjing 211106, China; ZOE110@163.com (J.T.); cmmi@nuaa.edu.cn (C.M.); 2School of Economics and Management, Nanjing Institute of Technology, Nanjing 211167, China; wxr920@163.com

**Keywords:** PM_10_ emissions, compactness, poly-centricity, vehicle mile travelled, congestion, Yangtze River Delta

## Abstract

The Yangtze River Delta (YRD) region is one of the most densely populated and economically developed areas in China, which provides an ideal environment with which to study the various strategies, such as compact and polycentric development advocated by researchers to reduce air pollution. Using the data of YRD cities from 2011–2017, the spatial durbin model (SDM) is presented to investigate how compactness (in terms of urban density, jobs-housing balance, and urban centralization) and poly-centricity (in terms of the number of centers and polycentric cluster) affect PM_10_ emissions. After controlling some variables, the results suggest that more jobs-housing-balanced and centralized compactness tends to decrease emissions, while poly-centricity by developing too many centers is expected to result in more pollutant emissions. The effect of high-density compactness is more controversial. In addition, for cities with more private car ownerships (>10 million within cities), enhancing the polycentric cluster by achieving a more balanced population distribution between the traditional centers and sub-centers could reduce emissions, whereas this mitigated emissions effect may be limited. The difference between our study and western studies suggests that the correlation between high-density compactness and air pollution vary with the specific characteristics and with spatial planning implications, as this paper concludes.

## 1. Introduction

The rapid urbanization and urban expansion have already undermined environmental improvement efforts, particularly in many urban areas across China where air pollutant emissions are increasing [1,2,3,4]. For national regulations, the Ministry of Environmental Protection (MEP) has required an air quality monitoring network that consists of 1436 monitoring stations in all prefecture-level cities since 2015, and these monitoring stations have begun to record hourly data for air pollutant emissions [5]. According to MEP [6], less than a quarter of Chinese cities reached the Chinese air quality standard (Air Quality Index < 100) in 2016. Furthermore, severe haze pollution has begun to affect eastern central areas in China [7]. Due to its severe public health impacts, the poor air quality in China has attracted high attention from the government and academic sources [8].

It is not difficult to identify major factors that ultimately lead to an increase in air pollution emissions by the urban transport sector. Longer travel distance and increasing dependence on private cars are all important factors that can immediately increase air pollutant emissions. Moreover, urban road networks carrying an increasingly large number of vehicles lead to an increasingly large amount of transport-related pollutant emissions to air, with the problem exacerbated by traffic congestion [9]. This transport-related pollutants’ problem is particularly relevant in China because the rapid increase in motorized private transport and the future automobile sales growth is projected to be 35% [10]. A significant part of air pollutant emissions originating from prolonging commuting on the road should be a focus for the application of strategies to reduce such emissions. 

To limit these emissions, many strategies originating from an academic source have been suggested for achieving a sustainable development of cities. Cities can adopt strategies such as promoting the provision of public transit [11,12] and steering the compact urban form and polycentricism indicated in recent studies. For instance, some empirical studies have focused on either European or US cities or Chinese cities by investigating the air pollution effects of different urban forms, in which the compact form was more likely to mitigate air pollution [8,13,14]. In addition, polycentric development with concentrating peripheral employment and population in sub-centers has been suggested to achieve lower levels of emissions [5,15]. Therefore, promoting compact urban form and polycentricism are the main strategies advocated by some researchers in order to reduce road traffic emissions, and, consequently, achieving sustainable development.

Many studies considered urban density as a primary factor explaining the differences in the level of air pollution [14,16,17,18,19,20]. Research on developed countries with a fairly long history of industrial and infrastructure development have provided rich evidence about the relationship between high-density compactness and air pollution. Consequently, empirical analysis for developing countries, such as China, is scarce. Recently, some studies investigated the relationship between urban form and air pollution in Chinese cities, in which the former has often been defined by several types of landscape variables, such as size, shape, regularity, fragmentation, and traffic coupling factor of urban patches by using satellite-derived imagery [5,21,22,23]. However, in the context of China, besides urban density, few have systematically been investigated at the metropolitan level despite how more nuanced but potentially important aspects of compact urban form, such as jobs-housing balance, and spatial structure, such as poly-centricity, impact air pollution. A more complete assessment on the importance of compactness and poly-centricity for air quality is necessary, as characteristics of urban form and spatial structure in Chinese cities are rather compact and polycentric. First, compared with developed countries, Chinese cities are considered to be more compact than American cities [24]. On the one hand, urban density in China is higher. For example, the average density in China is 6100 persons per square kilometer while the US and EU cities are 1200 and 3200 persons per square kilometer, respectively [10]. On the other hand, due to the existence of Danwei system [25,26], the jobs-housing balance is very high in Chinese cities compared to European or US cities [10,27]. The jobs-housing balance is very high in Chinese cities compared to European or US cities [10,27]. Second, with rapid growth of population in Chinese cities, decentralizing into the outskirts occurs and a polycentric development pattern emerges [28,29]. In practice, the recent example which reflects the new trend of polycentricism among Chinese cities is the construction of Xiong’An new district, which is a new city near Beijing [30].

Moreover, researchers suggested that compactness decreases air pollution, as compactness makes people drive less and, thus, less vehicle miles travelled (VMT) take place [16,18,19]. However, compact urban form with increasing concentrated origins and destinations pairs may also increase prolonging commuting time on the road, namely congestion [30], which is another important contributor to transport-related air pollutant emissions [9]. Accordingly, the impact of compactness on air pollution is determined by the net effect of the two countervailing forces. Similarly, following Ewing et al. [31] and Li et al. [30], we believe that the impact of poly-centricity on air pollution is also unclear. Polycentric development could reduce pollutant emissions for two reasons. First, poly-centricity facilitates proximity between housing and work in the clustered development of the suburbs, which reduces VMT [32]. Second, a polycentric city promotes commuting within or between sub-centers, in which decreasing concentrated travel flows in the main center [33] would reduce congestion. However, it should be notable that poly-centricity does not necessarily reduce both VMT and congestion in the context of Chinese cities. It cannot guarantee people living in a city’s sub-center do not commute to the city’s main center, since most high-quality resources such as amenities often remain concentrated in the main center of a polycentric city [34]. People living in the sub-centers tend to commute more and longer to consume high-quality resources of the traditional main center, which leads to an increase commuting distance and congestion [30]. Therefore, the net effects of compactness and poly-centricity on air pollution are, ultimately, an empirical question.

While a decrease in VMT and traffic congestion is likely to reduce air pollution, it should be noted that poor air quality does not necessarily result from longer VMT and prolonging commuting time on the road because two cities with a similar commute distance may still have different congestion levels or vice versa due to their differences in population size, private car ownerships, etc. Other factors such as car private ownerships and population size could also affect a city’s air pollution emissions and should be controlled when comparing air pollution emissions across different cities. In this regard, estimating the relationships between compactness, poly-centricity, and air pollution, which controls for such factors, is necessary for such comparative studies. 

Therefore, to fill these gaps, the Spatial Durbin Model (SDM) [35] is presented to estimate how the specific characteristics of Chinese cities have influenced the relation between compact and polycentric development and air pollution using the data of YRD cities from 2011–2017. We selected YRD cities in China as our sample because YRD is an area with one of the densest populations and the most developed economy in China. Air pollution is evaluated by the concentration of one main air pollutants: Particulate Matter (PM_10_), because the large number of PM_10_ emissions is produced by road transportation [14]. For example, according to Li et al. [36], road transportation in the whole YRD region contributed to considerable and significant PM_10_ emissions (i.e., 83.3 Gg) in 2010. Furthermore, we examine the heterogeneity that could potentially exist in the impacts, i.e., whether the influences would be heterogeneous for cities with a different number of vehicle ownerships. This paper is expected to facilitate understanding of the relationship between urban planning strategies aiming at influencing air quality in China, particularly in YRD cities. 

The following begins by describing the study area, data, and method. The following sections conclude empirical results and discussion. We then suggest policy implications and research limitations, which serve as departure points for future studies.

## 2. Study Area, Data, and Methods

### 2.1. Study Area and Data

#### 2.1.1. Study Area: YRD Area

The Yangtze River Delta (YRD) region consists of Shanghai, Jiangsu, Zhejiang, and Anhui provinces, and it is one of the most densely-populated and economically developed areas of China. Since the reform and opening up, urbanization and economic development have led to an unprecedented move of people from the countryside toward cities [37], especially YRD cities. The total population in this area experienced a rapid growth, increasing from 140.34 million in 2015 [38] to 190 million and ranked as one of the most developed and densest regions in the world [39]. Meanwhile, economically developed Eastern China had experienced increasing haze days [40]. For example, in 2015 and 2016, the average annual concentrations of PM_10_ of most of YRD cities were higher than the particulate pollutant emission limits of the United States and the European Union [41]. For the health impacts of PM_10_ emissions and the mitigation of such emissions, it is important and necessary to investigate whether or not the compact urban form and poly-centricity can improve air quality in YRD, which brings sustainable development. Therefore, our study area focuses on YRD cities. The boundaries selected for study are shown in Figure 1.

#### 2.1.2. Data 

The increasing number of vehicles (increased to 23.63 million in the YRD) [42] concentrated the road traffic flow. Additionally, long-range transport is generally considered a major contributor to this long-term increasing trend of PM_10_ concentration [43,44]. Such prolonging commuting on road will drastically increase road abrasion, which contributes to most of the locally emitted PM_10_ [44]. Despite ultrafine particles (<0.1µm) originating from increasing vehicle exhaust emissions, which also dominate the pollutant concentration, PM_10_ and ultrafine particles are well correlated due to the same source in traffic and possible differences in health impacts of different particle metrics that may not be readily seen long term [44]. In this paper, air pollution in terms of PM_10_ was used for measuring the level of air pollution. Accordingly, we stress that the impacts of urban form and spatial structure on increasing PM_10_ emissions are determined by both longer VMT and prolonging travel times between origins and destinations. 

Besides the magnitude measurement indicated in Reference [36], there is another concentration measurement of air pollutant emissions complied with those recommended in the China National Ambient Air Quality Standard [45]. Report on the State of the Environment in Chinese cities reports the data of air pollution concentration every year. We obtained data for the annual degree of PM_10_ concentration from it in 19 YRD cities from 2011 to 2017.

To measure poly-centricity and compactness, we use the data of residents, employments, and land size during the period of 2011–2017, which was obtained from the statistical yearbook of local cities (2012–2018). Recent papers on the urban spatial structure, in which more fine-grained cross-section population data was obtained in the LandScan™ High Resolution Global Population Dataset has been used to identify population centers. However, the shortcoming of this measurement is that inevitable errors might occur in estimating grid cell population [30]. Therefore, China’s statistical population data at the district level was used to help us identify population centers and measure spatial structure and urban areas over space and time following Engelfriet and Koomen [10] and Li et al. [30], which is the key step to the subsequent measure of poly-centricity and compactness. 

Selection of the time period is based on the availability of consistent data for early periods. The period selected for this paper is between 2011 and 2017, which covers seven years. The analysis methods are applied to these seven years. The reason to choose this period of time is that the consistent data of urban form and spatial structure are available for 19 cities only from 2011. 

### 2.2. Measuring Poly-Centricity and Compactness

The key characteristics of the urban development pattern are centralization or decentralization and concentrated or deconcentrated economic activity [46]. The centralization of the former refers to the degree employment is concentrated in a central business district (CBD), and the concentration of the latter represents poly-centricity [15], which suggests how employment is disproportionately clustered in some locations [47]. In this paper, we mainly combined two aspects of urban form and the spatial structure proposed by Muñiz and Garcia-López [15] and Li et al. [30]. We incorporated the jobs-housing balance into a measure of compactness in order to better understand the urban form and spatial structure in the Chinese cities. Therefore, urban forms can be high-density, jobs-housing-balanced and centralized compactness, and the spatial structure can be concentrated (polycentric) or decentralized (monocentric). Table 1 shows the definitions of compactness and poly-centricity variables examined in this paper.

#### 2.2.1. Measuring Poly-Centricity

Many early studies build on the developed-monocentric city model [48,49], which uses the assumption of employment concentrates in CBD and a surrounding residence concentrated area. Since employment is assumed to be concentrated in the center-CBD, commuting distances increase linearly with the distance from it. However, urban development processes do not follow the monocentric development in one main center. As agglomeration occurred, cities began sprawl. They started to take on a more polycentric form in which employment and population are clustered in sub-centers rather than that of the original center-CBD. During the clustered development in the outskirts, sub-centers emerged, which are increasingly independent from the traditional center-CBD [50]. Therefore, since the features of urban form can be polycentric or monocentric [51,52], many researchers tend to incorporate poly-centricity into the model and into the assessments. For example, the notable “co-location” hypothesis defines that people have more opportunities when they live closer to their work place, and a polycentric city reduces commuting distance [32,53].

Compared with a monocentric urban form, a polycentric form also tends to reduce congestion, which will reduce commuting time [52]. However, it can be argued that poly-centricity does not necessarily mitigate congestion. On the one hand, the reducing VMT effect may be dominated by increasing the congestion with high urban density [10]. On the other hand, the increasing trend of people living in the sub-center but consuming in the main center may concentrate traffic flows between sub-centers and the original center (CBD) [30], which may increase both VMT and congestion effects. 

Following Engelfriet and Koomen [10] and Li et al. [30], we focus on population distribution within the administrative boundary of urban districts (*Shiqu*), which usually form the core of Chinese cities development. Then, the procedure of the measurement of poly-centricity begins with the identification of activity centering in each city as follows. First, the number of activity centering (denoted by *DZN*) was identified. This study identifies population centers by a two-steps method, which is used in recent studies [10,54]. The definition of activity centering is based on three criteria: (i) the density in one activity centering is higher than the average of the city-wide mean density, (ii) adjacent polygons above the specified threshold are grouped together, and (iii) the activity centering must contain $(10/\sqrt{population})\%$ or more of the total city population. Second, the indicator of the polycentric cluster (denoted by *DZI*) is calculated.

The first indicator (*DZI*) of the polycentric cluster is calculated in the following method: DZI=N*HM*R, where N is the number of activity centering, R is the ratio of people who live in the identified activity centering to the total population size, and HM is the homogeneity index, which is calculated as follows. Based on the entropy index by Limtanakool et al. [55]: $HM=-\sum_{i=1}^{N}\left[S_{i}\ln(S_{i})\right]/\ln(N)$, where S_i_ is the ratio of the population who live in the activity centering to the population who live in all identified activity centering. The value of HM is between 0 and 1. If the population is equally distributed among all identified activity centering in a city equal to 1 and if all people concentrate on one activity, centering equals 0.

Then two variables were obtained to measure poly-centricity, including the number of activity centers and the degree of the polycentric cluster. The indicator-*DZN* defines the number of activity centers. A higher *DZN* suggests that a city has more activity centering and, hence, the city could be more polycentric. In addition, the indicator of the polycentric cluster is *DZI*, which suggests a high level of polycentric clustered city. The second indicator of the polycentric cluster (denoted by *SCS*) was from Reference [30], which reflects the population share of the city’s sub-centers to its centers. It indicates that the population is disproportionately distributed across the main center and sub-centers in a city. By definition, a higher value of *SCS* indicates a more polycentric-clustered city.

#### 2.2.2. Measuring Compact Urban Form

##### Urban Density

In European and American cities, the urban development pattern, which is considered to be a low urban density, produces automobile dependence and largely residential development along the periphery of the urban area [56]. There is a general consensus in studies that dispersion leads to longer commuting distances and more use of cars, as distances are generally longer [57]. Besides commuting distance, urban density can also affect commuting times, as many studies on US and European cities concluded that urban density is the main determinant of travel times [58,59]. Using the data of US cities, Levinson and Kumar [60] found that a high residential density (of 7500–10,000 persons per square mile) leads to the lowest commuting times. However, this feature of Chinese cities is not comparable to developed countries since Chinese cities are significantly denser than these developed urban areas, which poses the challenge for research findings found in developed countries that also hold for China [10].

There are many approaches to define urban density. We apply the most widely used measure of urban density (denoted by *PD*): resident density defined as the number of residential population in the district divided by its district area in a city and the city-level average population density was computed. Low resident density indicates urban sprawl, and high resident density suggests a compact urban form. Despite high-density compact cities making people drive less and have shorter VMT, the increasing congestion effect may negate the effect of reduction in pollutant emissions from decreasing vehicle usages. 

##### Jobs-Housing Balance

Besides urban density, poor accessibility can be found in dispersed development, where people must travel a long distance from one area to the other. Hamidi and Ewing [56] defines poor accessibility as single-use development, where segregation of and large private lots makes everything far apart. In sprawling urban areas, the segregated land use means that people have to travel long distances from the origin to destination, while the compact ones mean mixed land use because high accessibility makes everything more accessible. In this respect, balanced jobs-housing relation is expected to result in activities and reduce commuting distances and congestion [59]. Therefore, the jobs-housing balance tends to improve air quality in terms of PM_10_.

Following Hamidi and Ewing [56], this paper used one variable to measure the balance between jobs and population. Although using the same variable as Hamidi and Ewing [56] to operationalize mixed land use, the variables were computed differently by computing them within the boundaries of districts, because the population and employment data of a uniform one-mile buffer is unavailable in Chinese cities. The value of jobs-housing balance (denoted by *JBR*) is calculated by: (1)$$\mathrm{JBR}=\sum_{i=0}^{n}(1-\frac{ABS(J_{i}-JP*P_{i}}{J_{i}+JP*P_{i}})*\frac{BJ_{i}+BP_{i}}{TJ+TP}$$
where *i* is the district number, *n* is the number of districts in each city, *J* is jobs in the district, *P* is residents in the district, *JP* is jobs per person in the region, *BJ* is jobs in the district, *BP* is residents in the district, *TJ* is the total jobs in the region, and *TP* is the total residents in the region. By definition, a higher value of the jobs-housing imbalance defines dispersion (namely, low levels of compactness), which is the opposite of compactness. The increasing jobs-housing balance (*JBR)* is expected to result in shorter VMT [61] and higher travel frequency [62]. Therefore, the impact of the jobs-housing balance on PM_10_ concentration is determined by the net effect of these forces.

##### Urban Centralization

Based on Anas et al. [46], Lee and Gordon [47] developed the definition of urban spatial structures: the centralization, with development concentrated in the main center (CBD), and the concentration, with clustered development in sub-centers. In this paper, we mainly depend on the definition of centralization suggested by Li et al. [30] to better measure how disproportionately population distribute between its centers and non-centers. The centralization is represented by CBD, which indicates the share of the population of all centers in each city to its city-wide population. The high value of CBD means compactness, which is the opposite of sprawl. A high degree of centralization is expected to result in a reduction of VMT but increase congestion [30], which may offset the decreasing VMT effect.

### 2.3. The Regression Models

In order to fully undertake an assessment of urban compactness and poly-centricity’s air pollutant emission effect over the seven-year time period of this paper, ordinary least squares (OLS) regress models are used. The econometric model, presented in Equation (1), for a panel of i observations over t time periods is obtained by adding a subscript i, which runs from 1 to i (i.e., 19) and a subscript t, which runs from 2011 to t (i.e., 2017), to the variables and the error terms of the model. These models fit the traditional estimation of the impact of urban compactness and poly-centricity on air pollution. The natural form of the model is given as follows.
(2)$$\begin{array}{l} Ln(PM_{it})=\beta_{0}+\beta_{1}*Ln(PO_{it})+\beta_{2}Ln(COS_{it})+\beta_{3}Ln(JBR_{it})+\beta_{4}Ln(CBD_{it})\\ +\beta_{5}Ln(PD_{it})+\beta_{6}Ln(DZN_{it})+\beta_{7}Ln(DZI_{it})+\beta_{8}Ln(SCS_{it})+\mu_{i}+\rho_{t}+\varepsilon_{it} \end{array}$$
where *PM* represents a city’s annual levels of PM_10_ concentration, β_0_ is the constant term and β_n_ (*n* = 1, 2, ..., 8) is the coefficients, which are expected to be estimated, ε is the error term, μ is the cross-section effect of cities i, and ρ is time effect. Constant term in the linear regression is the intercept term, which is needed to be the estimated coefficient. The error term denotes the difference between the sample value and estimated value. More information about section and time effects could be found in Reference [35]. Besides dependent and independent variables in the model, there are control variables, which are included in it. The first variable suggests the size of the urban area [10]. It is represented by *PO*, which is the number of populations in a city. The second variable included in this paper is the number of vehicle ownership-*COS*. All control variables are collected or calculated from the Statistical Yearbook of YRD cities (2012–2018). This version of the model is estimated by using OLS and is used to estimate the impacts of compactness and poly-centricity on PM_10_. 

However, due to the violation of some basic assumptions of the regression model specification and invalidating the regression results in OLS models, Torrens [63] and Baumont et al. [64] have suggested the need to correct these errors. The suggested approach to correct errors of the OLS model, and accounting for spatial auto-correlation, is to use a spatial regression model, which adds a vector of weights of the dependent variable as an additional independent variable. The Spatial Durbin Model (SDM) then is used, which simply incorporates a vector of weights of the dependent variables as an additional independent variable into the OLS analysis. The results of SDM models are compared with that of OLS models for goodness of fit and other diagnostics on the coefficients. Basic form of SDM is:(3)$$Ln(PM_{it})=\lambda W*Ln(PM_{it})+\beta *Ln(X_{it})+\delta W*Ln(X_{it})+\mu_{i}+\rho_{t}+\varepsilon_{it}$$
where λ represents the spatial dependent effect, W represents the spatial weight matrix, X denotes both independent variables and control variables, δ denotes spatial spillover effects, μ indicates the cross-section effect of cities i, ρ denotes time effects t, and ε is the error term. λ, β, and δ are coefficients, which need to be estimated by using SDM. The spatial weights matrix (W) is derived by the equation and the data of all independent and control variables was from a statistical yearbook of YRD cities, as shown in Table 2. The coefficients of the independent variables, which do not represent the true partial regression coefficient, the total effect should be divided into a direct and an indirect effect. Referring to the method suggested by Elhorst [65], the SDM can be rewritten as the following:(4)$$Ln(PM_{t})={\left(I-\lambda W\right)}^{-1}{\left[\beta X_{t}+\delta W*Ln(X_{t})\right]}^{-1}+{\left(I-\lambda W\right)}^{-1}\varepsilon_{t}$$

The spatial differential equation matrix for the independent variable is as follows:(5)$${\left(\frac{\partial Ln(PM)}{\partial Ln(X_{ik})}\cdots \frac{\partial Ln(PM)}{\partial Ln(X_{nk})}\right)}_{t}={\left(I-\lambda W\right)}^{-1}\left(\begin{array}{cccc} \rho_{K} & W_{12}\rho_{K} & \cdots & W_{1N}\rho_{K}\\ W_{21}\rho_{K} & \rho_{K} & \cdots & W_{2N}\rho_{K}\\ \cdots & \cdots & \cdots & \cdots \\ W_{N1}\rho_{K} & W_{N2}\rho_{K} & \cdots & \rho_{K} \end{array}\right)$$
where the sum of the elements in the right matrix is the total effect, the mean of the diagonal element in the right matrix represents the direct effect, and the mean of the non-diagonal elements is the indirect effect (the spatial spillover effect). 

The spatial weight matrix (W) is exogenously given in the spatial econometric model. Therefore, a reasonable spatial weight matrix is a key step for the spatial econometric analysis. Generally, the contiguity-based weight matrix: if two locations that are geographically adjacent are 1, if they are not adjacent are 0, was often used in paper [66]. However, this adjacent effect cannot reflect a proximity effect of air pollution if these locations are close enough. Therefore, this paper used the geographic distance weight matrix to investigate the spatial effect of air pollution, since the PM_10_ emissions have the characteristic of a spatial distance configuration effect rather than a geographically adjacent effect. The matrix of spatial weights w_ij_ equals the inverse distance, including w_ij_ = 1/d_ij_ (i ≠ j), where d_ij_ refers to the physical distance between two cities (i ≠ j). If i = j, d_ii_ denotes the physical distance within cities, and its value is derived from $d_{ii}=(2/3)\sqrt{area/\pi }$, where the area is a city’s size. In Equations (3)–(5), where w_ij_ is the element at the ith row and jth column of the spatial weights matrix (W). The data of physical distance between cities was from the calculation of Baidu Map and the land size of a city was from the statistical yearbook of local cities (2012–2018). 

Due to a maximum likelihood (ML) being more efficient if the residual term follows normal distribution [67], in this analysis, the SDM is estimated by using ML estimation techniques. ArcGIS was used to generate the geographical distance weight matrix and run the Spatial Durbin Model (SDM) regressions. The effects of compactness and poly-centricity on PM_10_ are estimated using both OLS and SDM versions. Descriptive statistics about the variables for the 19 cities are listed in Table 2 and the correlations between explanatory variables are shown in Table 3.

## 3. Results

### 3.1. OLS and SDM Regressions Results for PM_10_

The OLS and SDM regressions results for PM_10_ are shown in Table 4. Table 4 presents the findings of the two regression models that describe the effect of compactness and poly-centricity on PM_10_. Although the results of OLS regression are nearly consistent with that of SDM regression, due to the OLS model shows and explains R^2^ to be around 0.19–0.28, we use the results of SDM regression as our base of analysis.

Table 4 reports how compactness and poly-centricity affect the level of PM_10_ emissions by considering control for variables, e.g., city size and private car ownerships. Interestingly enough, in Model 1, we find that city size (represented by population size-*PO*) has a significantly negative effect on PM_10_, which indicates that the larger city size tends to emit less PM_10_. One possible reason could be the limitation of OLS analysis and the consequent biased results. The impact of private car ownerships represented by *COS* is generally as expected. The coefficient of *COS* is positive and statistically significant at the 1% confidence level, which indicates that more private car ownerships in cities tend to experience more PM_10_ emissions, as expected.

To overcome the limitations of OLS regression models of (1)–(4), SDM models of (1a)–(4a) were estimated to overcome the presence of spatial auto-correlation. For the use of spatial econometric methods, spatial dependence is investigated first. The results show the values of local Moran’s Index. The Moran’s Index is usually used to measure the spatial dependence of variables. A positive value indicates a positive spatial dependence, while a negative value suggests the spatial auto correlation is negative. More information about Moran’s Index can be found in Reference [67]. (2011–2017) are around 0.323–0.451, which are statistically significant. It indicates that the concentration of PM_10_ has a significant-spatial correlation. This depends on the results of the Hausman test. Hausman test is a test for a selection of either a fixed effect model or a random effect model, of which it can be used for estimating the model. The fixed effect model is chosen if the result of the Hausman test passes the significant test, which indicates rejecting the original hypothesis [68], and vice versa. Models (1a)–(4a) did not pass the test of significance, and, therefore, the results of a random effect model were used for the final report of PM_10_. The full results of SDM are shown in Table 5.

### 3.2. SDM Results for PM_10_

In Table 5, the results indicate that the spatial coefficients of compactness and poly-centricity in Models (1a)–(4a) are consistent, which indicates that there is a significant spatial correlation and spatial spillover effect in the PM_10_ among YRD cities, and the results of SDM in Table 5 are used as the base of our analysis. Compared with the results of OLS models, the results of SDM models for city size (denoted by population size-*PO*) and private car ownerships (represented by *COS*) showed both a significant positive effect on PM_10_ emissions, which indicates that increasing population and vehicles’ ownerships have a considerable and significant effect on commuting distance and congestion. Specifically, the indirect and total effects of *PO* have a significant-positive effect on an increase in PM_10_ emissions, which indicates that the poor air quality of PM_10_ originates from the larger size of its neighboring and whole area. The one possible reason of an insignificant-direct effect may be that both increasing the jobs-housing balance effect [56] and the deterioration of the jobs-housing imbalance [10] exist in our sample due to the large size of a city, which results in both increasing and decreasing VMT and congestion effects. However, the considerable and significant increasing PM_10_ originated from its neighboring cities, which dominates the insignificant-direct effect. This leads to the whole PM_10_ emissions increasing. In addition, the direct, indirect, and total effects of private car ownerships represented by *COS* on PM_10_ is generally positive as expected, which indicates that more private car ownerships in cities have significant positive spatial correlation and spatial spillover effect on PM_10_ emissions.

Urban density denoted by *PD* has considerable and significant positive indirect and total effects on PM_10_, while its direct effect is positive but does not pass the significant test. It indicates that high PM_10_ emissions will be influenced by the high urban density of neighboring cities, while a high degree of emissions will be not be affected by its local effect. The insignificant-direct effect may be a result of population density that has no direct relationship with commuting distance and commuting time using OLS analysis, as shown in Engelfriet and Koomen [10]. However, the considerable and significant indirect effect originated from neighboring cities may dominate the insignificant-direct effect, and, thus, the total PM_10_ emission increases.

The direct, indirect, and total effect of jobs-housing balance (represented by *JBR*), which are −0.099, −0.054, and −0.153, respectively. All of the direct, indirect, and total effects of *JBR* pass the significant test. It indicates that more balanced jobs-housing relation can lead to less PM_10_ emissions, as shown in Cervero [69]. The significant impact may be a result of a jobs-housing balance that has resulted in less commuting distance, as well as the decrease of congestion. The indirect and totals effects of urban centralization denoted by *CBD* are significant-negative, which indicates that better air quality in terms of less PM_10_ emissions may depend on the degree of urban centralization from its neighboring cities. The high degree of centralized economic activity of neighboring cities has a large-spatial spillover effect on PM_10_ emissions. This considerable and significant decreasing air pollutant emission effect from neighboring cities may offset an increasing or decreasing direct emission effect of local cities, and, thus, lead to PM_10_ emission declines.

The positive signs of the indicator of poly-centricity (denoted by the number of activity centers-*DZN*) suggest that a more polycentric urban form tends to experience more PM_10_ emissions. All of the three effects (i.e., direct, indirect, and total effects) of developing many activity centers show significant and positive impact on PM_10_. It indicates that the increasing PM_10_ emissions are affected by developing too many activity centers that come from both local and neighboring cities. The one possible reason may be that increasing too many activity centers may also lead to more commuting activity [59,70,71,72] and use more private vehicles [73] to make increasing VMT and congestion effects [30]. PM_10_ emissions then increase. However, the positive signs of the other two indicators of poly-centric clusters (*DZ**I*, *SCS*) do not have a significant influence on PM_10_, which indicates that the ambiguous relationship between poly-centric clusters and commuting. Based on the case of US and European cities, studies on developed countries have shown both negative and positive correlations between the two. The fact that we are not able to establish a significant relation between poly-centricity (denoted by *DZI* and *SCS*) and PM_10_ emissions also suggests that the relations between poly-centricity and commuting (i.e., commuting distance and congestion) may depend on other local factors existing in our sample. 

Therefore, model (4a) investigates the heterogeneity that may exist in the impacts of polycentric clusters (measured by *SCS*) on PM_10_. When incorporating the term of the *SCS***COS* into model (4a), the insignificant-positive effect of poly-centricity on PM_10_ turns to be stronger and more significant. Let the coefficient of *SCS* (i.e., 0.46*ln (*COS*)-0.45) equal zero. The finding shows that the effect is negative for cities with private car vehicles of >10 million within cities. For such cities, polycentric-clustered development may improve air quality in terms of PM_10_. However, this improving effect may be limited, since the coefficients of three effects are around 0.3%−0.9%.

## 4. Discussion

For those YRD cities, the jobs-housing balance and urban centralization increase seems to be two good strategies to reduce air pollution in terms of PM_10_. Cities that are denser or develop too many activity centers tend to produce more air pollution emission while those with a more balanced jobs-housing relation and centralized economic activity tend to reduce air pollutant emission. Even though polycentric-clustered development tends to lead to less air pollution for cities with a larger number of private car ownerships (of >10 million), since this improved effect may be limited for its coefficient of the total effect, which is about 0.9%. 

The most surprising result is the coefficient of the urban density. Supporters of the compact development consider the main instrument to reduce air pollution to maintain or increase population density levels. Compared with the findings of Europe, the US, and China, the main difference is that we find a considerable and significant positive effect of urban density on air pollution. This result stems from estimating the SDM where all control variables and urban form and spatial structure variables are included to capture controversial relations. By incorporating either the number of activity centers indicator or polycentric cluster indicators, density is statistically significant. In contrast with many studies, the higher level of density exerts more PM_10_ emissions in this paper. This difference may be explained by two to five times higher density in Chinese cities, particularly in YRD cities, which may make a rapid increase in congestion since this may take over a decreasing VMT effect and, thus, air pollutant emissions increase.

In addition, since the literature on the impact of balanced-jobs-housing compactness on air pollution is lacking, we introduce the jobs-housing balance into the measure of a compact urban form and found that the jobs-housing balance has a significant and negative effect on PM_10_ emissions, which indicates that a more balanced jobs and housing relation may reduce air pollutant emission, as shown in Cervero [69]. In this respect, jobs and housing location, not density, is the spatial dimension that is an effective strategy for mitigating air pollutant emission occurring in YRD cities in China, as suggested by Glaeser and Khan [74]. 

Generally, our empirical evidence supports the recent trend of polycentric development of Chinese cities (particularly for these large cities) on the mitigation emissions effect. However, it also indicates that excess emphasis on polycentric development by developing too many activity centers may also lead to more air pollutant emissions. In contrast with the findings of Muñiz and Garcia-López [15] and She et al. [5], poly-centricity in terms of the number of centers tends to result in more PM_10_ emissions, which indicates developing too many centers that would lead to a dominant congestion effect. Moreover, it also finds that polycentric development by enhancing or reinforcing a more balanced population distribution between the traditional centers and sub-centers may also lead to less PM_10_ emissions, particularly for cities with large private car ownerships (>10 million within cities). However, this mitigated emissions effect may be limited, as this relationship may be determined by other local characteristics, such as the jobs-housing balance that we could not further investigate in this paper. 

In the context of China, especially in YRD cities, we would argue that appropriate long-term strategies for reducing pollutant emissions lie only in addressing the issue associated with jobs-housing-balanced and concentrated compactness, which underlies fewer PM_10_ emissions because it decreases both vehicle miles travelled (VMT) and congestion effects, while increased high-density and polycentric development by developing too many centers are not effective strategies for reducing such air pollutant emissions. However, for cities with a larger number of private car ownerships (>10 million within cities), enhancing polycentric clusters in terms of achieving a more balanced population distribution between the traditional centers and sub-centers could reduce emissions, whereas this mitigated emissions effect may be limited.

## 5. Conclusions

Urban form plays a key role in explaining the concentration of transport-related pollutant emissions. Regarding the rapid urbanization and urban expansion and the consequent long-lasting impacts on accessibility, housing affordability, and air pollutants, knowledge on the issue of how compactness and poly-centricity affect air quality is of crucial importance for China. For the mitigation of pollutant emissions, many studies suggest developing compact urban forms and polycentricism to reduce the level of emissions. However, the effect of compact form on air quality is usually controversial. In addition, knowledge on the effects of poly-centricity and compactness of an urban form in terms of jobs-housing balance on air quality in Chinese cities is scarce. 

The current study takes the first step toward a better understanding of the relationships between compact urban form and poly-centricity and air quality for a sample of 19 YRD cities by using the analysis of the spatial durbin model (SDM). Notwithstanding the limitations of OLS analysis, the SDM provide a unique opportunity to be able to investigate the spatial effect of pollutants and the way to mitigate emissions. The analysis indicates that air pollutants in these cities responds differently to some urban characteristics than was expected from the evidence in the case of the US and EU cities. 

The findings show that compactness in terms of urban density, jobs-housing balance, and urban centralization are important determinants for air pollutants in terms of PM_10_ emissions. The rapid urbanization and urban expansion have led to clustered development in the outskirts, as yet most of jobs and amenities remained concentrated in the traditional centers. Hence, the large effect of an increase in city size on VMT and congestion, cities with a larger city size tend to emit more PM_10_. In addition, cities with more vehicle ownerships have a large effect on PM_10_ emissions, which indicates that more vehicles used are one of the reasons for more PM_10_ emissions. After controlling these variables, cities with higher urban density have more PM_10_, while higher degree of jobs-housing balance and urban centralization have less PM_10_. In addition, poly-centricity in terms of developing too many activity centers has a higher level of PM_10_ emissions, while enhancing polycentric clusters does not necessarily result in more PM_10_ emissions. However, for cities with more private car vehicles of >10 million, polycentric clustered development may improve air quality in terms of PM_10_, but this improvement may be limited. Therefore, our paper indicates that jobs-housing balanced and concentrated compactness can be two effective strategies to reduce air pollution. 

Our findings suggest that strategies aimed to reduce air pollution by compactness and polycentric development may need to be reconsidered in the context of Chinese cities. According to the findings of this empirical study, policy implications for cities in developed countries should be cautiously developed. Some incomparable results of our empirical findings, such as the positive impact of compactness in terms of urban density on air pollutants in the context of YRD cities, might contribute to the longstanding debate and the impact of compact development on air pollution in developed countries. In addition, our empirical results also contribute additional evidence to the literature and provide some policy implications for Chinese cities that aim to mitigate air pollution by polycentric development. To some degree, our findings support the current polycentric development pattern in Chinese cities, as polycentric development enhances a more balanced population distribution between the main center and its sub-centers for those cities with a large number of private car ownerships (>10 million within cities). However, poly-centricity by developing too many centers may also lead to more air pollutant emissions.

However, the data used in this paper have limitations, which need to be further extended. First, following Engelfriet and Koomen [10] and Li et al. [30], we use population centers rather than employment centers to measure poly-centricity. Focusing on employment data will shed more light on the heterogeneity that potentially exists in the relationship between poly-centricity and air pollution, i.e., the impacts may be heterogeneous for cities with different jobs-housing balances, which is not considered in our paper. Second, the data obtained in this paper is limited to the administrative boundary of urban districts from the statistical yearbook of local cities. Future studies can use more a fine-grained level of data [56], to advance our understanding of compactness and poly-centricity in air pollution emissions.

## Figures and Tables

**Figure 1 ijerph-16-04204-f001:**
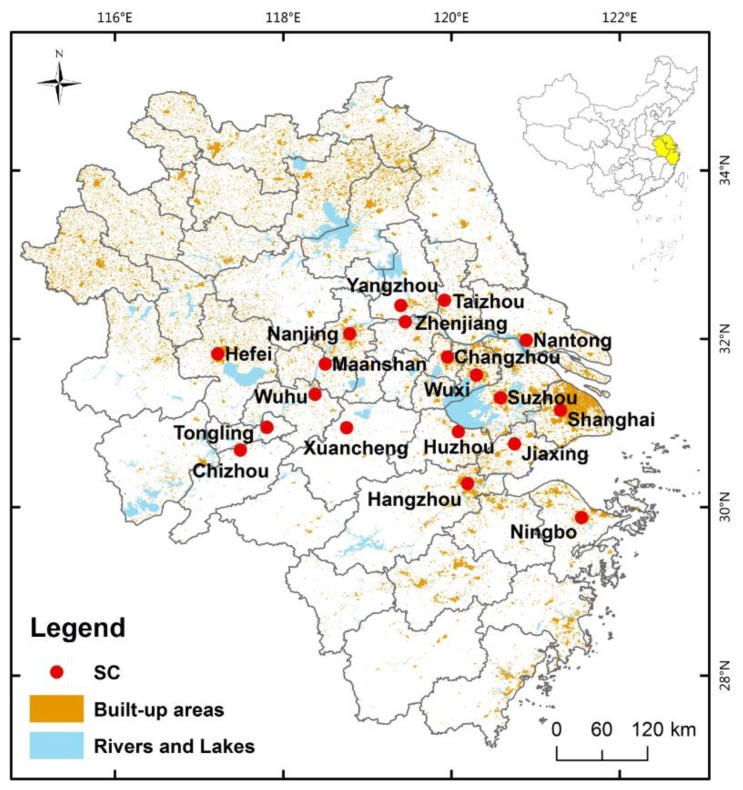
The location of Yangtze River Delta (YRD) and YRD cities (SC).

**Table 1 ijerph-16-04204-t001:** The descriptions of compactness and poly-centricity indicators.

Framework		Measures	Significance
Compactness	Urban Density ^a^	Average residential density (*PD*)	High average residential density suggests that a compact city
	Jobs-housing balance ^a^	Jobs-housing balance index (*JBR*)	High jobs-housing balance reflects a compact urban form
	Urban centralization ^a^	Centralized index (*CBD*)	High degree of urban centralization means that a compact city
Poly-centricity	Activity centers ^a^	The number of centers (*DZN*)	More centers suggest a polycentric city
	Polycentric cluster ^a^	Polycentric-clustered index (*DZI*)	High polycentric-clustered index reflects a polycentric city
		Population distribution between centers (*SCS*)	More balanced population distribution among centers reflects a polycentric city

^a^ The residents and jobs data were available at the district level. Hence, residents’ density, jobs-housing balance, urban centralization, the number of centers, and a polycentric cluster were assessed at the district level. In addition, due to the detailed district-level, jobs data was not available from the statistical yearbook of all YRD cities. Hence, after some revisions, we selected 19 YRD cities as our sampled cities and the boundaries of cities selected for study are shown in Figure 1.

**Table 2 ijerph-16-04204-t002:** Statistics of dependent, independent, and control variables.

Variables (Unit)	Minimum	Maximum	Mean	Standard Deviation	Data Sources
*PM* (mg/m^3^)	0.05	0.137	0.088	0.0163	Report on the State of the Environment of YRD cities
*PO* (ten thousand persons)	44.76	2425.68	402.9253	535.4233	The number of districts, residents, employments, private car ownerships at the district level and at the city level was from a statistical yearbook of YRD cities
*COS* (ten thousand vehicles)	4.079	201.5554	66.6280	55.5321	
*JBR* (-)	0.9674	9.8788	3.6055	2.0370	
*CBD* (%)	0.1176	1.1020	0.3898	0.1953	
*SCS* (%)	0	0.8131	0.2743	0.2796	
*DZN* (-)	1	8	2.0451	1.5888	
*DZI* (-)	0	3.3193	0.5931	0.7373	
*PD* (persons/km^2^)	263	10,004	1930	2065	

**Table 3 ijerph-16-04204-t003:** Correlations between independent and control variables.

	Ln *PO*	Ln *COS*	Ln *JBR*	Ln *CBD*	Ln *SCS*	Ln *DZN*	Ln *DZI*	Ln *PD*
Ln *PO*	1.0000							
Ln *COS*	0.8856	1.0000						
Ln *JBR*	0.6235	0.5200	1.0000					
Ln *CBD*	0.5069	0.4235	0.2820	1.0000				
Ln *SCS*	−0.6582	−0.5517	−0.4062	−0.0235	1.0000			
Ln *DZN*	0.4234	0.2987	0.1878	0.0663	−0.5827	1.0000		
Ln *DZI*	−0.3970	−0.3378	−0.3177	−0.1141	0.4776	0.2964	1.0000	
Ln *PD*	0.2682	0.1242	0.1330	0.2710	−0.2863	0.3192	−0.0318	1.0000

**Table 4 ijerph-16-04204-t004:** OLS and SDM regression results-PM_10._

Independent Variable			Dependent Variable (Natural log PM_10_)					
	OLS(1)	SDM(1a)	OLS(2)	SDM(2a)	OLS(3)	SDM(3a)	OLS(4)	SDM(4a)
Constant	9.0250 (1.68)*	18.1499 (3.99)***	9.8983 (1.80)*	20.0873 (3.95)**	19.9759 (3.22)***	27.900 (4.85)***	−25.0033 (−2.08)**	0.0742 (0.01)
*PO*	−0.3061 (−3.94)***	−0.4327 (−5.69)***	−0.1157 (−1.57)	−0.3379 (−3.62)***	−0.16667 (−1.63)	−0.4204 (−3.43)***	−0.0920 (−0.88)	−0.3639 (−3.91)***
*COS*	0.2162 (3.90)***	0.1272 (2.52)**	0.1505 (2.61)**	0.1027 (1.99)**	0.1629 (2.65)***	0.1116 (2.06)**	0.5671 (4.86)***	0.3446 (3.59)***
*PD*	−4.6379 (−2.56)**	−5.6953 (−10.24)***	−3.3747 (−1.87)*	−5.1400 (−4.94)***	−3.0788 (−1.58)	−5.2961 (−5.83)***	−3.9704 (−2.17)**	−5.6286 (−9.35)***
*JBR*	−0.0566 (−1.69)*	−0.0760 (−2.47)**	−0.0736 (−2.09)**	−0.0839 (−2.62)***	−0.0804 (−2.09)**	−0.0886 (−2.95)***	−0.0874 (−2.49)**	−0.0922 (−4.13)***
*CBD*	0.2105 (4.11)***	0.3285 (5.62)***	0.1496 (3.32)***	0.2982 (5.00)***	0.1636 (2.73)***	0.3334 (5.11)***	0.1352 (2.22)**	0.3144 (5.50)***
*DZN*	4.0240 (4.35)***	2.1791 (2.01)**						
*DZI*			0.1380 (2.73)***	0.0712 (1.25)				
*SCS*					0.0053 (0.06)	0.0440 (0.58)	0.6959 (3.75)***	0.3561 (2.69)***
*SCS***COS*							−0.0075 (−4.07)***	−0.0045 (−2.86)***
R^2^	0.2777	0.8289	0.2252	0.7520	0.1901	0.7905	0.2755	0.8601
N	133	133	133	133	133	133	133	133
LP		−507.8803		−510.1872		−511.3723		−516.0860
rho		2.0229 (7.01)***		2.0305 (6.61)***		2.0176 (6.47)***		1.9080 (5.08)***
Hausman effect		−55.07	−65.09					−329.321

****p* < 0.01, ***p* < 0.05, **p* < 0.1.

**Table 5 ijerph-16-04204-t005:** SDM Regression results-PM_10._

Independent Variable	Dependent Variable—PM_10_											
	Model (1a)			Model (2a)			Model (3a)			Model (4a)		
	Direct effect	Indirect effect	Total effect	Direct effect	Indirect effect	Total effect	Direct effect	Indirect effect	Total effect	Direct effect	Indirect effect	Total effect
*PO*	0.012 (0.09)	1.023 (2.98)***	1.036 (2.20)**	0.134 (1.14)	1.085 (3.27)***	1.219 (2.85)***	0.093 (0.67)	1.170 (3.79)***	1.263 (3.06)***	0.134 (1.16)	1.161 (3.79)***	1.295 (3.29)**
*COS*	0.165 (2.53)**	0.085 (1.94)*	0.250 (2.43)**	0.134 (1.95)*	0.072 (1.46)	0.207 (1.83)*	0.145 (2.05)**	0.076 (1.56)	0.221 (1.94)*	0.450 (2.62)***	0.244 (2.57)**	0.694 (3.40)**
*PD*	3.033 (1.19)	19.931 (3.07)***	22.964 (2.57)**	4.123 (1.61)	21.215 (3.21)***	25.338 (2.82)***	4.747 (1.81)	22.854 (3.36)***	27.601 (2.99)***	3.353 (1.48)	20.827 (3.42)***	24.180 (2.95)**
*JBR*	−0.099 (−2.41)**	−0.054 (−1.64)*	−0.153 (−2.14)**	−0.110 (−2.64)***	−0.061 (−1.65)*	−0.170 (−2.29)**	−0.116 (−2.77)***	−0.063 (−1.64)*	−0.178 (−2.33)**	−0.121 (−3.84)***	−0.068 (−2.32)**	−0.189 (−3.25)**
*CBD*	0.029 (0.36)	−0.694 (−2.79)***	−0.664 (−2.08)**	−0.020 (−0.26)	−0.738 (−3.10)***	−0.757 (−2.51)**	−0.006 (−0.08)	−0.779 (−3.89)***	−0.785 (−2.96)***	−0.010 (−0.14)	−0.761 (−3.66)***	−0.771 (−2.89)**
*DZN*	2.806 (1.95)*	1.419 (1.66)*	4.225 (1.92)**									
*DZI*				0.092 (1.31)	0.047 (1.13)	0.139 (1.28)						
*SCS*							0.059 (0.81)	0.028 (0.53)	0.087 (0.59)	0.460 (2.91)***	0.247 (2.40)**	0.707 (2.87)**
*SCS***COS*										−0.006 (−2.96)***	−0.003 (−2.31)**	−0.009 (−2.85)**

****p* < 0.01, ***p* < 0.05, **p* < 0.1.

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
