# Peer review of "Do Compactness and Poly-Centricity Mitigate PM10 Emissions? Evidence from Yangtze River Delta Area"

_ijerph, 2019, doi:10.3390/ijerph16214204_

Round 1

Reviewer 1 Report

Do compactness and poly-centricity mitigate PM10 2 emissions? Evidence from Yangtze River Delta area

Tao J et al.

This paper investigates how and poly-centricity affect air pollution in terms of PM10 in Yangtze River Delta region (China).

I see the topic of interest since it is an emerging area and the results could be applicable to other similar areas.

Although air pollutants generated by transport are more extensive (NOx, CO, VOC and PM2.5), it seems clear that PM10 as an indicator could be good since the existence in the air of a greater amount of PM10 means always worse air quality.

On the contrary, certain parts of the paper are very messy and its reading is sometimes difficult to understand and tedious.

In my opinion the paper could be published. However, there are a number of improvements that I believe necessary before it can be published.

Introduction

Introduction should briefly mention the main aim of the work. This paper starts at line 104 with “we investigate”, then line 107 “In addition…, and line 112 “Then…. Continues with line 115 We also …, paragraph 117-126 Second…,and paragraph 127-131 Third…It seems to be too messy. Consider to be more briefly and focus on the objectives of the study. The description of the works must be included in the second section (methods).

Paragraph 132-140 should not be in introduction either since it seems to be a discussion theme.

Line 32. MEP should be defined. It is defined later in line 177 (Ministry of Environmental Protection) but it has to be defined the first time it appears.

The same happens with Line 76 where VMT should be defined. It is defined also later in line 135 (Vehicle Mile Traveled) but it has to be defined the first time it appears.

Ref 33 is missing or at leats I could not find its cited. It goes from 32 (line 89) to 34 (line 122)

Methods

Some equations are numbered and other not. Consider to number all the equations and cite them first if it is possible.

Fig 1. Needs a complementary map to locate YRD in China. It would also be necessary to add a scale to get an idea of the size of the region.

Line 197. CBD should be defined. It is defined later in 207 (Central Business District) but it should be defined the first time it is mentioned.

Line 235 . “the indicator of polycentric cluster is calculated “. Consider to add (denoted by DZI) to make it more understandable.

Line 247. It seems to be two polycentric clusters indicators (DZI and SCS) but only DZI has been described and justified through an equation. SCS is been never named or justified first. We are not even able to know what that acronym means.

Line 286. Consider to place first JBR= ….. to make it more understandable.

Line 310. Number 8 is in bold font unnecessarily. Constant term, error term, section effect and time effect are not explained. May be adding a reference could help to understand them.

The same happens with Spatial Durbin Model (SDM) , the paper explains the meaning of the variables used but not how to get the values or where they come from. Adding a reference could help.

Line 362. Consider to add a page break to avoid cutting the table.

Results

Line 384 and 386. Talks about local Moran’s Index and Hausman test. They are never been explained before, so consider to add a new reference for both.

Line 384-385. Appears 2 “are” followed.” … are based, are around”. Rewrite the sentence to make it clearer.

Line 410. Appears the word un-significant. Insignificant or negligible are more common used.

Author Response

Dear Reviewer 1:

We are very thankful and excited to have been given the opportunity to revise our manuscript, titled “Do compactness and poly-centricity mitigate PM10 emissions? Evidence from Yangtze River Delta area” and ID is “ijerph-623826” for International Journal of Environmental Research and Public Health. We carefully considered the comments advised by your kind self. Those comments are all valuable and very helpful for revising and improving our paper, as well as the important guiding significance to our researches. We have studied comments carefully and have made correction which we hope meet with approval. For your kind consideration, revised parts are marked as red in the revised version of paper. The main corrections in the paper and responds to your valuable comments are as following:

Point 1: Lines 104-132 Introduction should briefly mention the main aim of the work.

Response 1: We have re-written this part according to your suggestion. We have improved this sentence as the main aim of this paper. (See lines 108-119)

Point 2: Paragraph 132-140 should not be in introduction either since it seems to be a discussion theme.

Response 2: Considering your suggestion, we have putted this part into the discussion section as shown in the revised version. (See lines 495-503)

Point 3: Line 32. MEP should be defined.

Response 3: We are very sorry for our negligence of the definition of MEP when it appeared for the first time. We have corrected it in the revised version. (See lines 32-33)

Point 4:  The same happens with Line 76 where VMT should be defined.

Response 4: We are so sorry again for our negligence of the definition of VMT when it appeared for the first time. We have already corrected it. (See line 83) 

Point 5: Ref 33 is missing.

Response 5: We have made a correction according to your comment. 

Point 6: Some equations are numbered and other not.

Response 6: We have made a correction according to your comment by adding the number of equation. (See line 264) 

Point 7: Fig 1. Needs a complementary map to locate YRD in China.

Response 7: As reviewer suggested that we have made an improvement of the map in China. We have added the location of YRD region in China map in Figure 1. (See lines 138-140)

Point 8: Line 197. CBD should be defined.

Response 8: We are so sorry again and again for our negligence of the definition of CBD when it appeared for the first time. We have already corrected it. (See line 175)

Point 9: Line 235. Consider to add “(denoted by DZI)” to make the sentence - “the indicator of polycentric cluster is calculated” more understandable.

Response 9: In line 212 in revised version, we have added “denoted by DZI” into that sentence. (See line 212) 

Point 10: Line 247. SCS is been never named or justified first.

Response 10: We are very sorry for our negligence of the definition of SCS for the first time. We have re-written the sentence in the revised version. (See lines 225-228)

Point 11: Line 286. Consider to place “JBR= …..” to make the sentence more understandable.

Response 11: As your suggestion, we have added “JBR=” into equation (1). (See line 264)

Point 12: Line 310. Number 8 is in bold font unnecessarily. Constant term, error term, section effect and time effect are not explained. May be adding a reference could help to understand them.

Response 12: Considering your suggestion, we have adding a reference of constant term, error term, section effect and time effect (See note 3 at the end of page 8). In addition, we have corrected the unneccessarily bold font in number 8 (See line 300).

Point 13: The same happens with Spatial Durbin Model (SDM), the paper explains the meaning of the variables used but not how to get the values or where they come from. Adding a reference could help.

Response 13: Considering your suggestion, we have adding a reference of SDM, including the meaning of the variables and how to get values and the sources of data. (See note 4 at the end of page 8)

Point 14: Line 362. Consider to add a page break to avoid cutting the table.

Response 14: We have corrected this problem in revised version.

Point 15: Line 384 and 386. Talks about local Moran’s Index and Hausman test.

Response 15: Considering your suggestion, we have adding a reference of local Moran’s Index and Hausman test for explaining the meaning of Moran’s Index and Hausman test. (See notes 5 and 6 at the end of page 10) 

Point 16: Line 384-385. Appears 2 “are” followed.” … are based, are around”. Rewrite the sentence to make it clearer.

Response 16: As suggested by the reviewer, the sentence has been referred to the revised manuscript to reveal the evidence of spatial auto-correlation by calculating Moran’s Index. (See lines 376-379)

Point 17: Line 410. Appears the word un-significant. Insignificant or negligible are more common used.

Response 17: After examining your comment carefully, we must admit that we have not used more common word in the previous manuscript. We have revised the word “un-significant” to “insignificant” in the revised version. (See lines 393, 397, 405, 408, and 438)

Once again, thank you very much for your comments and suggestions.

We tried our best to improve the manuscript and made some changes in the manuscript according to the reviewers’ comments. These changes will not influence the content and framework of the paper.

We appreciate for Editors/Reviewers’ warm work earnestly, and hope that the correction will meet with approval.

Once again, thank you very much for your comments and suggestions.

Yours Sincerely,

Reviewer 2 Report

This paper used spatial regressions to analyze the effect of the poly-centricity and compactness effect of PM10 concentrations (ug/m3) on Chinese cities of the Yangtze River Delta Region (YRD). The authors presented a sound scientific approach with a comparison with OLS and Spatial Durbin Model and suggesting that cities with less than 10M vehicles compactness can reduce emissions, while cities bigger than 10M polycentric approach could reduce as well. My recommendation is to accept this paper after correction of the following issues:

During the entire manuscript, the authors mention the relationship between city form and PM10 emissions, which are obtained from the Chinese Statistical Yearbook. However, Table 1 shows that the unit of PM10 us (ug/m3), which are concentration and not emissions. Emissions is a flow of mass time (Mt/year) for instance, while the concentration is the mass of pollutant in a volume, which is the atmosphere. As a consequence, the PM10 concentrations in YRD posses a seasonality due to the meteorology and emissions patterns. That said, it is correct to state that the city form affects the PM10 emissions because it affects VMT and travel times between origin and destinations and with higher emissions worst the air quality (as shown on line 175). 

The introduction is too long with many small paragraphs that could be merged in fewer ones. Also, there are some text paragraphs in section 2 that should be in the introduction, for instance, lines 176-180.

Section 2.2 shows some concepts and section 2.3 of the regression models. However, these sections need to improve clarity, for instance, the parameter SCS it is only shown on line 247 without any mention of the meaning of the letters SCS. As a resume, despite that these parameters are mentioned in the text, these sections need to improve clarity.

Line 70. Define Danwei system

Lines 106-107. Add a reference about sources of PM10 emissions using YRD regions, authors may check the MEIC emissions inventory (http://meicmodel.org/)

Line 117. Add reference for Spatial Dubin Model.

Fig 1. Improve the quality of the figure.

Table 2. I'm curious about the correlation of 0.65 between population SCS.

Author Response

Dear Reviewer 2:

We are very thankful and excited to have been given the opportunity to revise our manuscript, titled “Do compactness and poly-centricity mitigate PM10 emissions? Evidence from Yangtze River Delta area” and ID is “ijerph-623826” for International Journal of Environmental Research and Public Health. We carefully considered the comments advised by your kind self. Those comments are all valuable and very helpful for revising and improving our paper, as well as the important guiding significance to our researches. We have studied comments carefully and have made correction which we hope meet with approval. For your kind consideration, revised parts are marked as red in the revised version of paper. The main corrections in the paper and responds to your valuable comments are as following:

Point 1: It is correct to state that the city form affects the PM10 emissions because it affects VMT and travel times between origin and destinations and with higher emissions worst the air quality (as shown on line 175)

Response 1: As suggested by the reviewer, lines 150-152 in revised version have been stressed the importance of the influencing mechanism of urban form and spatial structure. (See lines 150-152) 

Point 2: Table 1 shows that the unit of PM10 us (ug/m3), which are concentration and not emissions.

Response 2: In the review comments, the reviewer has pointed out that “the unit of PM10 (ug/m3) are concentration and not emissions”. We have again examined Table 1 carefully, and have explained two measurements of air pollution in revised version, where included the magtitude measurement (e.g. Li et al., (2017)) and concentration measurement (e.g., Li et al., (2019)). We have used the data of PM10 concentration as our measurement of PM10 due to the availability of city-level data of PM10 concentration. (See lines 153-156)

Point 3: The introduction is too long with many small paragraphs that could be merged in fewer ones.

Response 3: As suggested by the reviewer, we have deleted some sentences and merged smaller paragraphs in original version, and have made more brief sentences in revised one. (See lines 108-119) 

Point 4:There are some text paragraphs in section 2 that should be in the introduction, for instance, lines 176-180.

Response 4: As suggested by the reviewer, we have added this sentence into introduction as shown in lines 32-35.

Point 5: These sections need to improve clarity, for instance, the parameter SCS it is only shown on line 247 without any mention of the meaning of the letters SCS.

Response 5: As suggested by the reviewer, we have explained the meaning of SCS in revised version. (See lines 225-228)

Point 6: Line 70. Define Danwei system

Response 6: We are very sorry for our negligence of the definition of Danwei system when it appeared for the first time. We have already added a note of the definition of Danwei system. (see note 1 at the end of page 2)

Point 7: Lines 106-107. Add a reference about sources of PM10 emissions using YRD regions, authors may check the MEIC emissions inventory (http://meicmodel.org/)

Response 7: As suggested by the reviewer, we have already added a reference about sources of PM10 emissions in YRD region. (See lines 114-115)

Point 8: Line 117. Add reference for Spatial Dubin Model.

Response 8: As suggested by the reviewer, we have already added a reference about Spatial Durbin Model. (See note 2 at the end of page 3)

Point 9: Fig 1. Improve the quality of the figure.

Response 9: In the review comments, the reviewer has pointed out that “improve the quality of Figure 1”. We have made an improvement of the quality of Figure 1, such as adding the location of YRD region in China map and making the boundaries of cities more detailed. (See lines 138-140, Figure 1)

Point 10: Table 2. I'm curious about the correlation of 0.65 between population and SCS.

Response 10: As the reviewer suggested, we examinied Table 2 in the previous version carefully, we found that the coefficient of population and SCS exerted the relative-high correlation between the two. It indicates that the potential multi-coefficient problem in the model. Here we presented some results by using different variables and different models (i.e., OLS and SDM analysis) for testing the robustness of our results as shown in the following Table. More information can be found in the attachment. 

Reviewer 3 Report

The manuscript aims at discussing if compactness of poly-centricity lead to lower PM10 emissions with as case study the Yangtze River Delta in China. The topic fits the journal but it is not very clear how novel it is as the overall outcome is as expected. Most critically though, the manuscript is very non-transparent about the data it uses. There is no clear data source referenced for many kinds of data. Other numbers come also out of the blue as they do not match the references cited. Finally some data appear to have wrong units All this leads to zero confidence in the validity of what is presented. Data needs to be far more transparent and it is completely unacceptable to cite papers to support data (data that is nto in these papers). I can only urge rejection.

Examples of problems (not complete list)

“For example, the average density in China is 6100 persons per square kilometer while the US and EU cities are 1200 and 6100, respectively [9].” In Reference 9 there is no info on these 3 numbers! This is unacceptable. You need to clearly explain where numbers come form Datasets do not seem to match: 2011-2017 vs 2012-2018? Why different periods? “he selection of 19 cities in YRD has been a challenging exercise. The main reason was that it is difficult to access to the same data in all YRD cities, only the data of 19 YRD cities were available from the data sources “ What data was used>? Where form? Links or citations of reports (if the data is actually given in the reports). It should be some form of accessible database. What precautions were taken that this does not introduce a bias (smaller cities excluded vs larger ones) Table 1: units do not make sense? So you have an average population density of 400 million people per km square? So all of China fits on 2 km2? Nonsense like this leaves no confidence in any of your data and data handling.

Also the manuscript needs English polishing throughout

Define abbreviations at first use (e.g. MEP)

What is the Danwei system?

Special symbols are all mis-formatted  (um)

Figure 1. Map resolution is poor

Author Response

Dear Reviewer 3:

We are very thankful and excited to have been given the opportunity to revise our manuscript, titled “Do compactness and poly-centricity mitigate PM10 emissions? Evidence from Yangtze River Delta area” and ID is “ijerph-623826” for International Journal of Environmental Research and Public Health. We carefully considered the comments advised by your kind self. Those comments are all valuable and very helpful for revising and improving our paper, as well as the important guiding significance to our researches. We have studied comments carefully and have made correction which we hope meet with approval. For your kind consideration, revised parts are marked as red in the revised version of paper. The main corrections in the paper and responds to your valuable comments are as following:

Point 1: There is no clear data source referenced for many kinds of data.

Response 1: After examining the reviewer’s comments carefully, we admit that we have made a mistake and we were not expressed our data sources clearly in the previous manuscript. We are really sorry for our mistake. In the revised version, we have clearly explained data sources. These are displayed in Table 2. (See lines 349-350)

Point 2: In reference 9 there is no information on these 3 numbers.

Response 2: In the review comments, the reviewer has pointed out that “In reference 9 there is no information on these 3 numbers”. We have carefully examined the numbers appeared in the previous version, we are so sorry for our typo mistakes between line 69 and line 71. In the revised version of paper, we have made a correction in the light of your kind suggestion (See lines 74-76). In addition, the figures can be found in the study of Engelfriet and Koomen (2018) (the link of this study can refer to https://core.ac.uk/download/pdf/81763427.pdf; and the detailed information can see in lines 7-9 of page 1271 from top).  

Point 3: Some data appear to have wrong units.

Response 3: After examining the previous version carefully, we must admit that we have made a typo mistake when we are writing. We are very sorry for our negligence of the importance of unit of data. We have already corrected the unit of data in Table 2 in revised version (See Table 2. line 349-350).

Point 4: You need to clearly explain where numbers come from Datasets do not seem to match: 2011-2017 vs 2012-2018?

Response 4: After examining the reviewer’s comments carefully, we must admit that we have not expressed our meaning correctly in the previous manuscript. Sorry for this confusion. In the revised version, we explained that the study period is from 2011-2017. The data was gathered from different sources. Specifically, the data source of PM10 concentration was from Report on the State of the Environment in YRD cities (2011-2017) and other data was obtained from statistical yearbook in YRD cities (2012-2018), the statistical yearbook reports the data of previous year such as data of 2011-2017 in report of 2012-2018, (See lines 158-160).

Point 5: The selection of 19 cities in YRD has been a challenging exercise. The main reason was that it is difficult to access to the same data in all YRD cities, only the data of 19 YRD cities were available from the data sources “ What data was used>? Where from? Links or citations of reports (if the data is actually given in the reports).

Response 5: We must admit that we have not expressed our meaning correctly in the previous manuscript. We have deleted this sentence, “The main reason was that it is difficult to access to the same data in all YRD cities, only the data of 19 YRD cities were available from the data sources” and added clear data sources in Table 2. 

Here we can offer links of statistical yearbooks of samplied cities. Specifically, the number of districts, employments, residents and land size at the district level and city level can refer to http://221.226.86.104.ipv6.nanjing.gov.cn/file/index.htm

The same data of Shanghai can refer to http://tjj.sh.gov.cn/html/sjfb/tjnj/

The data of Wuxi can refer to http://tj.wuxi.gov.cn/ztzl/tjnj/index.shtml

The data of Changzhou can refer to http://tjj.changzhou.gov.cn/class/OEJCMFCP

The data of Suzhou can refer to http://www.suzhou.gov.cn/xxgk/gmjjhshfztjxx/tjnj/

The data of Nantong can refer to http://tjj.nantong.gov.cn/

The data of Yangzhou can refer to http://tjj.yangzhou.gov.cn/yztjj/sdzkw/tjj_list.shtml

The data of Zhenjiang can refer to http://tjj.zhenjiang.gov.cn/

The data of Taizhou can refer to

http://tjj.taizhou.gov.cn/art/2016/6/4/art_2443_591843.html

The data of Hangzhou can refer to http://www.hangzhou.gov.cn/col/col805867/

The data of Ningbo can refer to http://tjj.ningbo.gov.cn/col/col18616/index.html

The data of Jiaxing can refer to http://tjj.jiaxing.gov.cn/col/col1512382/index.html

The data of Huzhou can refer to http://tjj.huzhou.gov.cn/tjsj/tjnj/index.html

The data of Hefei can refer to http://tjj.hefei.gov.cn/8688/8689/18nj/

The data of Wuhu can refer to http://tjj.wuhu.gov.cn/tjnj/tjnj.htm

The data of Maanshan can refer to http://tjj.mas.gov.cn/4702501.html

The data of Tongling can refer to http://www.tl.gov.cn/sjtl/tjnj/

The data of Chizhou can refer to http://sjcz.chizhou.gov.cn/tjnj/index.htm

The data of Xuancheng can refer to

http://tjj.xuancheng.gov.cn/content/channel/5a4afc4f20f7fedf28e8fa8e/

Unlike the above data sources, the data of PM10 concentration was from Report on the State of the Environment of YRD cities. For example, in Nanjing, from 2011- 2017, the data of PM10 concentration was from Report on the State of the Environment, and the specific data of PM10 concentration can refer to

http://hbj.nanjing.gov.cn/njshjbhj/201806/t20180605_1325808.html

The same data of Shanghai can refer to  

http://sthj.sh.gov.cn/sh/list_new.jsp?channelid=2149

The data of Wuxi can refer to

http://bee.wuxi.gov.cn/zfxxgk/xxgkml/index.shtml?ChannelID=1202

The data of Suzhou can refer to

http://www.szhbj.gov.cn/hbj/xxgkhbj/044003/044003001/

The data of Nantong can refer to http://hbj.nantong.gov.cn/ntshbj/hjzkgb/hjzkgb.html

The data of Yangzhou can refer to

http://hbj.yangzhou.gov.cn/yzhbjceshi/ndhjzlgb/list.shtml

The data of Zhenjiang can refer to http://hbj.zhenjiang.gov.cn/zwgk/hbgb/

The data of Taizhou can refer to http://hbj.taizhou.gov.cn/col/col43871/index.html

The data of Hangzhou can refer to

http://epb.hangzhou.gov.cn/col/col1692349/index.html

The data of Ningbo can refer to http://sthjj.ningbo.gov.cn/col/col16699/index.html

The data of Huzhou can refer to

https://www.aqistudy.cn/historydata/monthdata.php?city=%E6%B9%96%E5%B7%9E

The data of Jiaxing can refer to

http://www.jiaxing.gov.cn/art/2018/6/15/art_1555291_26762214.html

The data of Hefei can refer to

https://www.aqistudy.cn/historydata/monthdata.php?city=%E6%B9%96%E5%B7%9E

The data of Wuhu can refer to

http://sthjj.wuhu.gov.cn/NewsList.aspx?pTypeID=11030201

The data of Maanshan can refer to http://tjj.mas.gov.cn/4702501.html

The data of Tongling can refer to http://sthjj.tl.gov.cn/5906/5940/5942/5945/

The data of Chizhou can refer to

http://sthjj.chizhou.gov.cn/list.aspx?MenuID=003001001

The data of Xuancheng can refer to

www.xuancheng.gov.cn/opennessSearch/?type=2&keywords=%E5%85%AC%E6%8A%A5&field=title&branch_id=&from_date=&to_date=&button=%E6%9F%A5%E8%AF%A2

The data of Changzhou can refer to http://sthjj.changzhou.gov.cn/class/HQPMBJFE

Point 6: What precautions were taken that this does not introduce a bias (smaller cities excluded vs larger ones)

Response 6: With due respect, in our opinion the bias within the analysis can be neglected due to the following reasons. First, rapid urbanization and urban expansion in Chinese cities have resulted in many environmental problems, including the poor air quality. This issue is particularly important for large Chinese cities, because residents longer commuting distance, more dependence on private car usages and the increasing traffic congestion on road are all main factors which can immediately increase air pollution. Moreover, using the data of 30 large Chinese cities, recent study (i.e., Engelfriet and Koomen, 2018) investigated the impact of urban form on commuting and found that current sprawled urban development pattern may have long and negative consequences for accessbility in large Chinese cities. Following this study, we believe that air pollutant emssions by transport sector in large Chinese cities are more considerable and significant because the large number of population concentrates in large cities and rapid increase in private car useages, thus it is important and necessary to activate effective strategies to mitigate such air pollution in the large cities. In contrast, air pollutant emissions by tansport sector in small cities which are characterized as smaller size, less number of industries, lower population, and lower burden of traffic can be neglected. Therefore, we selected YRD region as our sample due to it is the most developed and populated region in China which can be considered as a strong representative sample when investigating the environmental impacts of compact urban form and poly-centricity and we believe that our analysis doesn’t pose any bias.

Second, we showed the robustness of our results in the paper. For example, after we incorporated different polycentric variables (i.e., poly-centricity in terms of DZN, DZI and SCS) into SDM, the signs and significances of PD, JBR, CBD exert consistent results (such as the results of SDM (1a), SDM (2a) and SDM (3a) in Table 5). In addition, we selected two indicators of polycentric cluster, which also exert the consistent signs and significances in Table 5, indicating the robust results of polycentric cluster (i.e., the results of model (2a) and model (3a) in Table 5). More information can be found in the attachment.

Point 7: Define abbreviations at first use (e.g. MEP)

Response 7: We are very sorry for our negligence of the definition of MEP when it appeared for the first time. According to another reviewers suggestion, we have added the same sentence into the instruction in revised version. (See Lines 32- 33)

Point 8: What is the Danwei system?

Response 8: In the comments, the reviewer has pointed out that “What is the Danwei system”. On page 2 of the revised manuscript we have clearly defined Danwei system. (See note 1 at the end of page 2)

Point 9: Special symbols are all mis-formatted (um)

Response 9: After examining the special symbols appeared in original version carefully, we have corrected all mis-formatted symbols in revised version. (See line 349-350, Table 2)

Point 10: Figure 1. Map resolution is poor

Response 10: As reviewer suggested that we have made an improvement of the map, such as adding the location of YRD region in China map and making the boundaries of cities more detailed in Figure 1 (See lines 139-140).

Point 11: The manuscript needs English polishing throughout

Response 11: We have made careful examination in the original manuscript and we also have consulted native English speakers for paper revision before the submission this time. We have already corrected the grammar mistakes appeared in original version and used more common term in revised one.

Once again, thank you very much for your comments and suggestions.

We tried our best to improve the manuscript and made some changes in the manuscript according to the reviewers’ comments. These changes will not influence the content and framework of the paper.

We appreciate for Editors/Reviewers’ warm work earnestly, and hope that the correction will meet with approval.

Once again, thank you very much for your comments and suggestions.

Yours Sincerely,

Round 2

Reviewer 3 Report

The manuscript is clearly improved and the sources of the data are somewhat clearer, still it leaves a really bad after taste that any serious data issues were just "typos". This does not inspire much confidence in this work.

Also the manuscript still does not read smoothly.